# Left Ventricular Hypertrophy and Ventricular Tachyarrhythmia: The Role of Biomarkers

**DOI:** 10.3390/ijms24043881

**Published:** 2023-02-15

**Authors:** Ljuba Bacharova, Marta Kollarova, Branislav Bezak, Allan Bohm

**Affiliations:** 1International Laser Center CVTI, 841 04 Bratislava, Slovakia; 2Premedix Academy, 811 02 Bratislava, Slovakia; 3National Institute for Cardiovascular Diseases, 833 08 Bratislava, Slovakia; 4Faculty of Medicine, Comenius University in Bratislava, 831 01 Bratislava, Slovakia

**Keywords:** left ventricular hypertrophy, ventricular tachyarrhythmias, biomarkers

## Abstract

Left ventricular hypertrophy (LVH) refers to a complex rebuilding of the left ventricle that can gradually lead to serious complications—heart failure and life-threatening ventricular arrhythmias. LVH is defined as an increase in the size of the left ventricle (i.e., anatomically), therefore the basic diagnosis detecting the increase in the LV size is the domain of imaging methods such as echocardiography and cardiac magnetic resonance. However, to evaluate the functional status indicating the gradual deterioration of the left ventricular myocardium, additional methods are available approaching the complex process of hypertrophic remodeling. The novel molecular and genetic biomarkers provide insights on the underlying processes, representing a potential basis for targeted therapy. This review summarizes the spectrum of the main biomarkers employed in the LVH valuation.

## 1. Introduction

Left ventricular hypertrophy (LVH) detected by ECG and imaging methods is an important cardiovascular risk factor associated with increased mortality and morbidity [1,2], the most dangerous life-threatening complications in LVH being ventricular tachyarrhythmia/fibrillation and sudden cardiac death.

LVH is a relatively general definition. The basic definition defines hypertrophy as the enlargement of an organ or its parts due to the enlargement of its components (as opposed to hyperplasia, i.e., the increased number of the components). For LVH, it means the enlargement of the left ventricle or its parts due to the enlargement of the cardiomyocytes. However, the etiology of LVH varies considerably including LVH in hypertensive disease and valvular diseases, cardiomyopathies, but also LVH in athletes, obesity, metabolic syndrome, myocardial infarction, amyloidosis, Chagas disease, etc.

The term “biomarker” is also a rather broad term. In biomedical contexts, a biomarker, or biological marker, is a measurable indicator of some biological state or condition. Currently, it is used as a more specific term for molecular or genetic structures identifying particular physiological or pathological processes. The changes in the myocardium during LVH development are very complex: they include inflammation, fibrosis, and apoptosis, affecting electrogenesis and creating a substrate for arrhythmia. Biomarkers can be characteristic of these processes, representing circulating signaling molecules initiating processes affecting the ventricular myocardium and triggering VF.

The individual biomarkers describe specific aspects of the complex hypertrophic re-building of the left ventricle. Each of these biomarkers gives a partial view on LVH, and it cannot be expected that a single parameter can completely describe LVH and identify the risk of developing ventricular arrhythmias. The aim of this paper was to review the current biomarkers in terms of their broader meaning and their possibilities for diagnosing LVH and estimating the risk of ventricular arrhythmias.

## 2. Left Ventricular Hypertrophy

The spectrum of clinical conditions associated with LVH is wide. The most common cause of LVH is hypertensive disease. The prevalence of LVH ranges from 20% in mildly hypertensive patients to almost 100% in those with severe or complicated HT [3]. Valvular diseases represent another group of clinical conditions leading to LVH due to pressure and/or volume overload. LVH can also develop in a response to work overload in healthy trained athletes, so-called “physiological” LVH.

Hypertrophic cardiomyopathy is an autosomal dominant genetic disease that is caused by a mutation in sarcomere protein genes, with a prevalence of 0.2% in the general population, according to echocardiographic studies [4]. The myocardium is characteristic with considerable disarrangement of myocardial tissue, fibrotic scars, and abnormal internal coronary vessels [5].

Myocardial infarction, especially the large and transmural, can be associated with local hypertrophy of the non-infarcted areas of the ventricles due to secondary volume-overload, as a part of post-infarction remodeling [6].

In obesity, the increase in LVM is conditioned not only by the growth of cardiomyocytes, but also by interstitial fat infiltration as well as the triglyceride accumulation in cardiomyocytes [7]. Similarly, the increase in LVM in amyloidosis is conditioned by the accumulation of amyloid in the myocardium [8]. Chagas disease is a major public health problem caused by *Trypanosoma cruzi*, with an estimated 6–7 million people infected and 70 million at risk of infection. Chagas cardiopathy is associated with left ventricular dilation and hypertrophy, its major complication being ventricular fibrillation and heart failure [9].

## 3. Methods of LVH Detection

The clinical methods for detecting LVH and the potential risk of developing ventricular arrhythmias are basically approaching this problem from three mutually related perspectives: (1) the imaging methods allow for the measurement of the size and structure of the left ventricle as well as some functional parameters such as contractility and hemodynamic parameters; (2) ECG records the electrical impulse generation and propagation; and (3) specific biomarkers identify the activity of the underlying processes at the molecular and genetic levels (Figure 1).

### 3.1. Imagining Methods in LVH Detection

In clinical practice, the diagnosis of LVH is based mostly on ECG and imaging methods: echocardiography and cardiac magnetic resonance (CMR). Since LVH is defined as the increase in left ventricular mass/size, the imaging methods can image the dimensions of LV and consequently estimate the left ventricular mass (LV mass), and thus they are naturally used as reference methods for LVH detection. They also provide additional important information on the contractility, hemodynamics, and tissue structure of the ventricular myocardium. Of special importance is the possibility of the late gadolinium enhancement MRI (LGE-MRI) to assess the ventricular tissue structure (i.e., to identify diffuse or localized electrically inactive areas of the myocardium).

#### 3.1.1. LVH Detected by Echocardiography

Based on the ultrasound beam reflection from the cardiac structures, echocardiography allows for noninvasive measurements of the dimensions of the left ventricle, and consequently to estimate the LV myocardial volume. The LVM is then calculated as a product of the estimated LV myocardial volume and the myocardial density of 1.05 g/mL [10]. The preciseness of the LVM estimation depends on the type of echocardiography used.

*M-mode echocardiography* represents the linear measurement: The M-mode echocardiography estimates the LVM using a rather rough formula, where the volume of the LV is calculated by a cubic formula and multiplied by the myocardial mass constant related to the myocardial tissue density. The geometrical assumption is that LV is a cube, which is the main limitation of this LVM estimation. The correlation between 2D-guided M-mode derived LVM and the postmortem LVM data is r = 0.78 [11]. This means that the coefficient of determination r^2^ = 0.61 indicates that 61% of results were in agreement with the postmortem LVM data and 39% was overestimated.

*2D-echocardiography*: The LVM estimation is based on the area-length and truncated ellipsoid methods, so the geometrical assumption is that the LV shape is an ellipsoid. The performance of 2D echo in estimating LVM compared to the autopsy findings has been shown to be comparable with 2D-guided M-mode echo (r = 0.66–0.72) [11]. This means that the values of the coefficient of determination were r^2^ = 0.43 and 0.52, respectively, meaning that about half of the echocardiographic results did not agree. The left ventricle size depends on the anthropologic parameter, therefore LVM indexed to the body surface area (LVM/BSA) is recommended for LVH detection. The diagnostic criteria recommended for 2D echocardiography are as follows: LVM/BSA [g/m^2^] ≥ 115 g/m^2^ in men and ≥95 g/m^2^ in women [10]. It has been shown that the LVM values are overestimated by M-mode [12,13], or either overestimated [13] or underestimated [14,15] compared to CMR.

*3D-echocardiography* estimates the LV volume without geometric assumptions, and thus the estimates of the LV volume are more precise. It showed considerably better agreement with CMR measurements compared to M-mode and 2D echocardiography (r = 0.99 vs. r = 0.84) [14].

In spite of the limitation, using echocardiography is cogent since they estimate the LV size, which correlates well with the LV mass measured with the reference methods such as autopsy and CMR. The more that by the definition LVH is defined as the increase in the size.

It was shown that LVH detected by echocardiography is significantly associated with ventricular arrhythmia and sudden cardiac death in patients with cardiomyopathy [15,16]. Furthermore, the left ventricular strain is associated with non-sustained ventricular tachycardia [17].

#### 3.1.2. LVH Detected by CMR

CMR is a nuclear magnetic resonance technique utilizing the phenomenon of the resonance of atomic nuclei in response to radiofrequency waves. For LVM estimation, it uses the short axis multislice (multiple 2D or 3D) cine acquisition, sampling the ventricles from the atrioventricular ring to the apex. The required volumes are derived with subsequent planimetry of the endocardial and epicardial borders of the ventricles. In this way, the derived volumes are independent of the geometric assumptions, which is a major advantage compared to 1D and 2D echocardiographic techniques [18]. To differentiate LVH, the reference values for adults are recommended as LVM 66–176 g for men, 41–125 g for women; LVM/BSA 30–85 g/m^2^ for men and 30–68 g/m^2^ for women. For more details, see [19].

The advantage of CMR is the use of advanced techniques for detailed morphological assessment of the myocardial tissue such as late gadolinium enhancement, parametric mapping, diffusion tensor imaging, and myocardial strain [20]. Thus, in patients with LVH, CMR provides a comprehensive and detailed evaluation of the degree and distribution of hypertrophy, ventricular function, and tissue morphological characterization, which are potentially promising as methods differentiating the etiology of LVH [20].

An important added value of CMR in LVH diagnosis is the method of the late gadolinium enhancement, which allows for the quantification of the extracellular volume—fibrosis. It has been documented that the increased proportion of the extracellular volume indicating fibrosis is significantly associated with ventricular tachyarrhythmia, a SCD in patients with hypertrophic cardiomyopathy [21,22,23].

### 3.2. Electrocardiography in LVH Detection

Electrocardiography diagnoses LVH as a dichotomous variable—LVH present or not. The basic diagnostic criterion of ECG-LVH is the increased QRS complex voltage. The classical diagnostic paradigm of ECG-LVH postulates that the increased LV mass generates a stronger electric field, which is reflected in the increased QRS amplitude of the surface ECG. Over the years, a considerable number of ECG-LVH criteria based on the increased QRS complex amplitude have been recommended [24]. The most commonly used ECG criteria for left ventricular hypertrophy are presented in Table 1.

However, only a minority of patients with an increase LVM also have increased QRS complex voltage, which is reflected in the low sensitivity of ECG-LVH criteria [29]. The discrepancies between the ECG and imaging methods results are considerable, and because the imaging methods can estimate the LV size, these methods are preferred for LVH diagnosis, and ECG is underestimated or neglected.

In relation to ventricular arrhythmias, this is quite interesting, since ECG is the only method recording the electrical activity of the heart, and thus the altered ventricular depolarization and repolarization are directly related to arrhythmias.

It also needs to be stressed that ECG in principle cannot measure the LV size—ECG records the distribution of the electrical potential on the body surface, and depends not only on the LV size, but mainly on the electrical properties of the myocardium, which are considerably altered in LVH [30]. The current trend in electrocardiology aims to shift the diagnostic paradigm of LVH and to re-focus the interpretation of the QRS complex changes from the size of the LVH on the electrophysiological underlying processes [31].

The presence of ECG-LVH criteria represents a risk of the higher incidence of ventricular arrhythmia and sudden cardiac death [32]. However, ECG parameters predicting ventricular arrhythmias are not limited to the increased QRS amplitude. A number of parameters of atrial depolarization, ventricular depolarization, and repolarization have been documented to be associated with ventricular arrhythmia/sudden cardiac death as independent and incremental risk factors including P wave abnormalities, increased QRS complex duration, prolonged QTc [33,34], fragmented QRS complex [34], and T wave abnormalities [35]. Interestingly, some of these parameters are included in the Romhilt–Estes score for ECG-LVH diagnosis [28]. Interestingly, also in obesity, which is often accompanied by LVH, the same spectrum of ECG abnormalities was identified as markers of risk for sudden death: leftward shift of P wave, QRS, T wave, P wave morphology, low QRS voltage, ECG-LVH, especially Cornell voltage and products, T wave flattening, and prolonged QT and QTc interval [36].

The predictive value of echocardiographic, CMR, and ECG LVH indicators for the development of ventricular arrhythmias/sudden cardiac death are presented in Table 2.

### 3.3. LVH by Molecular or Genetic Biomarkers

Cardiac and vascular function as well as structural abnormalities of the heart are reflected in a variety of different molecular or genetic biomarkers (biomarkers). Several factors influence the concentration of cardiac biomarkers including the presence of LVH [45,46,47,48], and they can signal adverse remodeling in patients with LVH to clinical heart failure. The importance of biomarkers lies in risk prediction and in potentially targeted therapy [49]. Some of them are already used in clinical decision-making—the “classic” biomarkers. The research is now focused on new promising biomarkers and their implications for diagnostics and focused therapy.

#### 3.3.1. Classic Biomarkers

##### N-Terminal B-Type Natriuretic Peptide (NT-proBNP)

B-type natriuretic peptides (BNPs) are produced primarily in the atrial and ventricular myocardium and circulate in the plasma. BNP induces vessel dilation and natriuresis, reduces preload and afterload, and consequently the myocardial stress. Distention and stretching of the LV wall leads to increased synthesis of BNP [50]. NT-proBNP circulates longer in the blood compared to active BNP and is eliminated through the kidneys. NT-proBNP is an independent prognostic marker for the risk of LVH in patients without HF [51]. The increased circulating levels of NT-proBNP predict the sustained ventricular tachycardia and may serve as an additional criterion defining patients at high risk for sudden cardiac death (SCD) [52]. It provides both diagnostic and prognostic information in the blood samples collected after the out-of-hospital resuscitation of patients with VF [53].

The increased levels of NT-proBNP seem to be associated with the increased occurrence of ventricular arrhythmias and SCD in patients with HF due to both ischemic and non-ischemic etiology [54]. Regarding ECG, the association between the increased levels of BNP and prolonged QTc was observed. It is speculated that these changes reflect the prolonged action potential duration that could result in electrophysiological abnormalities and ventricular tachyarrhythmias [55].

##### High-Sensitivity Cardiac Troponin I (Hs-cTnI) and High-Sensitivity Cardiac Troponin T (hs-cTnT)

The troponin (Tn) complex includes three subunits: TnT, TnI, and TnC. Cardiac troponins (cTnT, cTnI) are biomarkers of myocardial injury, mainly released during necrotic processes caused by myocardial ischemia. Several studies have also investigated the levels of Tns as potential markers in the risk assessment of malignant arrhythmias. Assays for high-sensitivity (hs) Tns are more sensitive and allow for the detection of lower concentrations [56].

hs-cTnT values are positively correlated with LVH in hypertensive patients [57]. It was reported that 78% of the patients with essential hypertension had increased values of hs-cTnT; additionally, these values were significantly related to age, glomerular filtration rate, and Cornell voltage criteria. The clinical importance of this biomarker in patients without coronary artery disease was also stressed [58]. Liu et al. [59] studied the association between levels of cTnI and ventricular arrhythmias in patients with chronic HF. VF was more likely to develop in severely decompensated HF patients with positive cTnI (>0.5 ng/mL) compared to patients with negative troponin.

##### Interleukin-6 (IL-6)

IL-6 is a cytokine, a small signaling protein with pro-inflammatory properties, that has an important role in the acute phase response. IL-6 is produced by macrophages in atherosclerotic plaques and is released by visceral adipose tissue and sub-endothelials. IL-6 increases the C-reactive protein (CRP) levels and is one of the molecules involved in the initiation of the inflammatory cascade [60].

In the study by Zhao et al., IL-6 deletion attenuated LVH and dysfunction, these findings indicating a critical role of IL-6 in the pathogenesis of LVH in response to pressure overload [61]. The association between IL-6 and ventricular arrhythmias was also observed in patients with established coronary artery disease [62].

##### C-Reactive Protein (CRP)

CRP is an inflammatory acute-phase protein, and its levels increase in injury or infection. In humans, CRP is mainly produced in the liver followed by smooth muscle cells of the aorta as well as by adipose tissue, and its production is mainly mediated by increased levels of IL-6 [63]. Elevated CRP levels are associated with LVH, indicating inflammation as a part of the complex processes of LVH progression.

CRP stimulates the absorption of low-density lipoprotein in the macrophages of endothelial cells. Furthermore, it contributes to atherosclerotic plaque progression and its conversion to unstable plaque. This may lead to coronary plaque rupture with the following ventricular arrhythmias and consequent SCD [64].

Elevated CRP levels were also observed in patients with torsades de pointes tachycardia and malignant arrhythmias. Interestingly, the CRP levels correlated with QT-interval prolongation. It is supposed that inflammatory cytokines might influence ion channel function with consequent alteration of the QT interval [65]. It has been shown that patients with structural heart disease experiencing electrical storms have higher levels of hsCRP, but also other biomarkers including IL-6 and NT-proBNP [66] compared to patients with single episodes of VT/VF or without ICD intervention.

#### 3.3.2. Novel Biomarkers

##### Galectin-3

Galectin-3 represents a link between inflammatory and fibrotic processes that are present in various cardiac pathophysiologies [67]. Galectin-3 is active on both the intracellular and the extracellular levels. Increased galectin-3 levels correlate with the degree of LVH in hypertrophic cardiomyopathy patients [68]. Galectin-3 is not a critical modulator of cardiac fibrosis, however, it may delay the subsequent hypertrophic response [69]. Erdogan et al. [70] focused on the association between levels of galectin-3 and the history of ventricular arrhythmias in patients with ischemic dilated cardiomyopathy with an implantable cardioverter-defibrillator. In this population, galectin-3 may be used for the risk stratification of patients who are more likely to develop life-threatening arrhythmias [70].

##### Derivatives of Reactive Oxidative Metabolites (DROM)

Inflammation and oxidative injury directly affect atrial myocyte contraction [71], electrical conduction [72], myocyte apoptosis, and cardiac fibrosis. Markers of oxidative stress have been identified in atrial tissue and appear to be associated with both inflammation and AF [72,73]; DROM levels also correlate with plasma CRP.

##### Matrix Metalloproteinases (MMPs) and Tissue Inhibitor of MMPs (TIMP)

MMPs are proteolytic enzymes involved in the degradation and remodeling of extracellular matrix under physiological and pathological conditions. MMPs also have regulatory and signaling functions [74] including inflammatory signaling. The upregulation of pro-inflammatory cytokines leads to increased MMP activation. However, long-term stimulation leads to increased levels of TIMP [75], followed by the decrease in the MMP/TIMP ratio, resulting in ongoing long-term remodeling. The ratio of MMP-9/TIMP-1 predicted the occurrence of tachyarrhythmias that required intervention, so the MMP/TIMP ratio shows a high potential for future applications [76]. MMPs reflect cardiac turnover processes with consequent remodeling, and could be a useful predictor of ventricular arrhythmias.

Elevated serum levels of MMP-7 were associated with the structural remodeling of LV in patients with LVH [77]. Lu et al. [78] hypothesized that MMP-3 polymorphisms associated with adverse myocardial remodeling could be associated with ECG changes due to increased collagen synthesis and disruption of efficient electrical conduction. They observed an association between MMP-3 genotype 5A/6A polymorphisms and QTc that was independent of age, gender, consumption of alcohol, smoking, BMI, or BP [78].

##### Apelin

Apelin is an endogenous peptide that can be detected in many tissues, endothelium, and human plasma, and acts as a ligand for the G-protein coupled APJ receptor. This system has a wide range of effects on the cardiovascular system including the modulation of cardiac contractility, vasomotor tone, renin–angiotensin system, cardiovascular development and repair, and many others. Apelin has also been studied as a potential candidate for the prevention of postischemic I/R injury, apoptosis, fibrosis, and remodeling and the treatment of heart failure.

Because of its role in many pathophysiological and physiological processes including oxidative stress and inflammation, apelin has been studied as a potential biomarker for several diseases including diabetes [79], atrial fibrillation [80,81,82], cancer [83], liver disease [84], and many others.

In terms of LVH, low apelin levels have been shown to be associated with LVH in untreated hypertensive patients, probably due to the AngII-mediated increase in cell size, protein content, and the expression of pro-hypertrophic and/or pro-fibrotic factors including TGF-b, OPN, and ANP [85].

Plasma apelin was also identified as an independent predictor of myocardial fibrosis in patients with hypertrophic cardiomyopathy [86]. These findings are in line with published data on the important role of apelin in cardiac apoptosis, fibrosis and remodeling, oxidative stress, and inflammation, which are all critical in the initiation and progression of ventricular hypertrophy [87].

A study by Ivankova et al. on hypertensive patients showed lower concentrations of apelin-13 in patients with ventricular arrhythmias [88,89]. Evidence based on in vitro animal models also shows that apelin shortens the duration of action potential in atrial myocytes via its effect on multiple ionic channels and directly effects the propagation of action potential and contractility in cardiomyocytes [90,91]. It is not clear whether apelin is involved in the etiopathogenesis of VA only as a result of fibrosis and LVH, or whether apelin also acts directly on the conductive myocardium, and by changing its electrical properties can contribute to the etiopathogenesis of VA.

However, despite these results, the role of apelin as a biomarker for LVH and VA remains unclear and merits further research.

##### Soluble Suppression of Tumourigenicity-2 (sST2)

ST2 is part of the IL-1 receptor family. It is related to myocardial dysfunction, fibrosis, and remodeling. Soluble ST2 (sST2) is also related to mechanical stress of the heart with consequent cardiac damage [87]. sST2 is a marker of myocardial stretch [92], while transforming growth factor-β1 (TGFβ1) plays a key role in the development of replacement fibrosis and myxomatous mitral valve degeneration [93]. Increased sST2 levels are associated with ventricular arrhythmias in patients with arrhythmogenic and hypertrophic cardiomyopathy [94].

Changes in biomarkers have also been observed in athletes exposed to intense, long-term strength, and endurance training. They had persistently elevated biomarkers, which led to the activation of multiple mechanisms including myocardial hypertrophy with subsequent fibrosis, ventricular enlargement, increased oxidative stress, or myocardial inflammation [95]. Aengevaeren et al. [96] observed that sST2 concentrations did not differ between athletes and patients with chronic or acute heart failure at the start line and after the marathon.

#### 3.3.3. Potential Biomarkers

##### Cardiotrophin-1 (CT-1)

Protein CT-1 is a member of the IL-6 family that signals via leukemia inhibitory factor receptor gp130-dependent pathways. CT-1 was originally characterized as a factor inducing cardiomyocyte growth and survival [97,98]. It has an important role in promoting changes in myocardial structure, and in the progression of LV remodeling. This remodeling results in LV failure in various cardiac diseases such as hypertensive heart disease, aortic stenosis, coronary artery disease, or dilated cardiomyopathy [99]. Plasma CT-1 levels were shown to be associated with the severity of LVH in patients with hypertrophic cardiomyopathy [100].

##### Growth Differentiation Factor 15 (GDF-15)

GDF-15 is a biomarker of cellular aging and systemic inflammation. Under physiological conditions, GDF-15 is not expressed in the heart. Expression increases rapidly in response to cardiovascular injury (i.e., pressure overload, ischemia). GDF-15 values were higher in hypertensive patients compared to patients with hypertrophic cardiomyopathy (HCM), and GDF-15 was an independent predictor of hypertensive LVH (H-LVH) in patients with LVH [101]. However, BNP values were lower in the H-LVH group compared to HCM group in this study. In untreated hypertensive patients, GDF-15 was related to increased LV mass index (MI) [102].

##### Annexin A5 (ANXA5)

ANXA5 is a 35 kDa protein that is a part of a family of calcium-dependent phospholipid binding proteins usually detectable in plasma. ANXA5 is one of the most abundant annexins in rat and human myocardium [103,104], which has a high affinity for phosphatidylserine. ANXA5 has anti-inflammatory and anti-apoptotic properties. ANXA5 was significantly associated with LVH in hypertensive patients, likely via influencing ANXA5 expression in serum and in myocardial cells [105]. In animal experiment, ANXA5 treatment attenuated the post-ischemic inflammatory response and ameliorated LV remodeling. This led to an improvement in cardiac function after MI-R injury in hypercholesterolemic mice [106]. The upregulation of myocardial AnxA5 is associated with the impairment of left ventricular systolic function in hypertensive heart disease (HDD) patients. Plasma AnxA5 may be useful as a biomarker of systolic dysfunction in patients with hypertensive heart disease [107].

##### Serum MicroRNA-27b (miR-27b)

MicroRNAs (miRNAs) are endogenous, single-stranded, short non-coding RNAs that act as regulators of gene expression by promoting the degradation or inhibiting the translation of target mRNAs. miRNAs play a fundamental role in diverse processes including cell development, differentiation, proliferation, and apoptosis [108]. Furthermore, they are important factors in cardiac hypertrophy and dysfunction. MiRNAs can be identified in human tissue, serum, and plasma. They are stable, have abundant circulation, and have a relatively easy methodology for detection, extraction, and quantification. This makes miRNAs one of the possible effective clinical biomarkers [109,110].

miRNAs are considered to have an important role in the development of cardiac hypertrophy [111]. The miRNA-27 (miR-27) family influences many cellular processes, and the beta isoform functions as an angiogenic switch by promoting endothelial tip cell fate and sprouting [112]. MiR-27b is frequently upregulated in pressure-overloaded hypertrophic hearts. Serum miR-27b is elevated in hypertensive patients with LVH [110], and Zhang and coworkers [113] assumed that circulating miR-27b is a possible specific noninvasive biomarker in screening for LVH patients. MiR-27b may also have a protective effect against cardiac dysfunction and hypertrophy by decreasing the expression of galectin-3.

##### Midregional Pro-Atrial Natriuretic Peptide (MR-proANP)

The natriuretic peptides (ANP and BNP) represent the gold standard of biomarkers in HF [114], and myocardial hypertrophy also leads to increased ventricular production of ANP and BNP [115]. These peptides have been established as valuable diagnostic markers and useful parameters that are able to assess disease severity and prognosis [116]. MR-proANP levels reflect the severity of HF, indicate the risk of cardiovascular events, and determine the time from the onset of AF to its manifestation [114].

Biomarkers are well-established in clinical practice for the detection, risk stratification, and monitoring of disease progression in various CVDs such as HF(NT-proBNP) or cardiac injury (troponins). Biomarkers of cardiac and non-cardiac origin might help to predict the risk of ventricular cardiac arrhythmias. “Classic” biomarkers of inflammation (CRP, hsCRP, IL6) have been studied in this regard, mainly due to the well-known relationship with chronic inflammation and have been shown to improve diagnostic efficiency in coronary artery disease, sudden cardiac death (SCD), or HF. Driven by these encouraging results and intensive research, an increasingly large number of “novel biomarkers” have emerged in recent years. Recent studies have described the significance of serum biomarkers as risk factors for ventricular tachyarrhythmias. However, although extensively studied, there is currently not enough evidence to use these biomarkers as a method for patient selection (e.g., for implantation of ICD) in the setting of LVH or VA and should be used only in context with other diagnostic tools [117,118].

## 4. Conclusions

Left ventricular hypertrophy is defined as an increase in left ventricular size, therefore, the primary diagnosis is naturally based on imaging methods estimating the left ventricular dimensions/mass. Currently, echocardiography and cardiac magnetic resonance are the dominant non-invasive methods in clinical practice for LVH detection. The role of ECG in LVH is now being re-evaluated and is moving away from estimating the LV mass to assessing the electrophysiological properties of hypertrophied myocardium [26]. The novel molecular and genetic biomarkers provide detailed information on the underlying processes in LVH and thus contribute to understanding the mechanisms of ventricular arrhythmia development. Identification of the underlying processes in LVH and their link to ECG and imaging methods remains a challenging but important clinical problem, with significant therapeutic and prognostic implications.

## Figures and Tables

**Figure 1 ijms-24-03881-f001:**
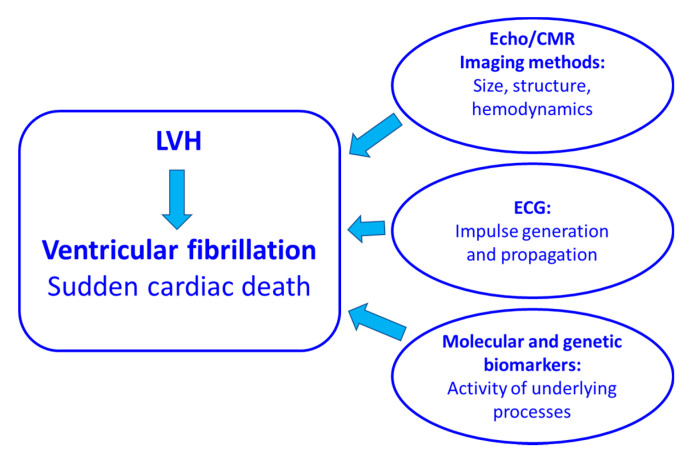
A spectrum of biomarkers providing information on the size of the left ventricle, its structure, hemodynamics, electrical impulse generation and propagation as well as on the activity of the underlying processes is currently available. They describe specific aspects of the complex hypertrophic rebuilding of the left ventricle. CMR: cardiac magnetic resonance, LVH: left ventricular hypertrophy.

**Table 1 ijms-24-03881-t001:** The most commonly used ECG criteria for left ventricular hypertrophy.

Criterion	Formula	LVH Criteria
Sokolow Lyon index [25]	SV1 + RV5 or V6	≥35 mm data
Cornell Voltage Criteria [26]	SV3 + RaVL	>28 mm (men)
	SV3 + RaVL	>20 mm (women)
Cornell Voltage-Duration Product [27]	(RaVL + SV3) × QRSd (men)	≥244.0 mVms
	(RaVL + SV3 + 0.8 mV) × QRSd (women)	
Romhilt–Este Score [28]	Amplitude of largest R or S in limb leads ≥ 20 mm (3 points)	≥5 points: definite LVH
	Amplitude of S in V1 or V2 ≥30 mm (3 points)	
	Amplitude of R in V5 or V6 ≥30 mm (3 points)	
	ST and T wave changes opposite QRS without digoxin (3 points)	
	Left Atrial Enlargement (3 points)	
	Left Axis Deviation (2 points)	
	QRS duration ≥ 90 ms (1 point)	
	Intrinsicoid deflection in V5 or V6 > 50 ms (1 point)	4 points: probable LVH

**Table 2 ijms-24-03881-t002:** The predictive value of echocardiographic, CMR, and ECG LVH indicators for the development of ventricular arrhythmias/sudden cardiac death.

LVH Indicator	Study Population	Risk Prediction
Echo: LVM [37]	SUD	OR 2.7, 95% CI 1.5–4.9; *p* = 0.001
CMR: Presence of LGE [23]	HCM	OR 2.52, 95% CI 1.4–4.4; *p* = 0.001
CMR: Presence of LGE [38]	HCM	OR 3.40, 95% CI 1.9–6.1; *p* < 0.001
CMR: Presence of LGE [39]	HCM	HR 10.01, 95% CI 1.2–83.8; *p* = 0.033
CMR: Presence of LGE [40]	HCM	OR: 3.41; 95% CI:1.97–5.94; *p* < 0.001
CMR: Presence of LGE [41]	HCM	HR: 1.08; 95% CI: 1.04–1.12; *p* < 0.001
CMR: extent of LGE (+10%) [40]	HCM	HR: 1.56; 95% CI: 1.33–1.82; *p* < 0.0001
ECG: SLI [37]	General population	OR 2.5, 95% CI 1.1–6.0; *p* = 0.04
ECG: SLI per mm increase [33]	Hypertension	HR; 95% CI 1.02 1.00–1.03; *p* < 0.030
ECG: CVDP per 100 mm.ms increase [33]	Hypertension	HR1.02; 95% CI 1.01–1.03; *p* < 0.001
ECG: SLI and Cornell voltage [42]	General population	HR 1.82, 95% CI 1.20–2.70; *p* = 0.006
ECG: Romhilt–Estes score [43]	Sudden cardiac arrest	OR 2.04, 95% CI 1.16–3.59; *p* = 0.013
ECG: QRSd per 10 ms increase [33]	Hypertension	HR: 1.26, 95% CI 1.18–1.34; *p* < 0.001
ECG: QTc ≥ 490 ms and T inversions [44]	General population	HR: 2.4; 95% CI: 1.2–4.9; *p* = 0.014 (women)
ECG: LBBB [33]	Hypertension	HR 3.24; 95% CI 2.19–4.81; *p* < 0.001

CMR: cardiac magnet resonance; LVM: left ventricular mass; LGE: late gadolinium enhancement; SLI: Sokolow–Lyon Index; CVDP: Cornell Voltage-Duration Product; QRSd: QRS duration; LBBB: left bundle branch block; SUD: sudden unexpected death; HCM: hypertrophic cardiomyopathy; OR: odds ratio; HR: hazard ratio.

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
