# Peer review of "Left Ventricular Hypertrophy and Ventricular Tachyarrhythmia: The Role of Biomarkers"

_ijms, 2023, doi:10.3390/ijms24043881_

Round 1

Reviewer 1 Report

The review article fits within the scope of the special issue Hunting for Prospective Molecular Approaches to Prevent Cardiac Arrhythmias. The title "Left ventricular hypertrophy and ventricular tachyarrhythmia: the role of biomarkers" clearly states the review's focus and specifies that the emphasis will be on the role of biomarkers in the relationship between left ventricular hypertrophy (LVH) and ventricular tachyarrhythmia.

The introduction is well-written and provides a sufficient background for the topic. It is an overview of left ventricular hypertrophy (LVH), its potential complications, as well as the role of biomarkers in diagnosing and predicting the risk of LVH and ventricular arrhythmias. The introduction also mentions the complexity of the processes involved in LVH development and the various clinical conditions associated with LVH diagnosis.

The conclusion lacks clarity due to the insufficient analysis of the biomarkers, mainly regarding their relations with ventricular arrhythmias.

There are major comments throughout the manuscript.

The review partially overlaps with the article “Circulating biomarkers in the early detection of hypertensive heart disease: usefulness in the developing world” DOI: 10.21037/cdt.2019.09.10. The specificity to biomarkers of arrhythmias in hypertrophied ventricles should be emphasized.

A table with diagnostic criteria of left ventricular hypertrophy should be included.

Prognostic assessments of the biomarkers should be included in one or more tables. One should be specific for ventricular arrhythmia prognosis (HR or OR when possible).

Figure one is not precise and/or does not reflect the content of the review. Please improve it.

Hypertrophic cardiomyopathy could be described better and dipper or relevant and recent guidelines cited. Chagas disease and amyloidosis are mentioned, but both should be further analyzed.

The left ventricular mass index should be included in the imaging section because it is a parameter used in echocardiography and cardiac MRI and is European and American guidelines. The differences in the cut points in different populations and by sex could also be mentioned.

The imaging techniques and the ECG are not the main markers described in the review, but the indication of the individual and/or combined contribution to the diagnosis and prognosis of ventricular arrhythmias should be included.

It should be indicated how much the diagnostic or prognostic improves by blood circulating biomarkers for those currently in use. If the contribution is not known yet, it must be clearly stated.

The following biomarkers are related to ventricular hypertrophy but not to ventricular arrhythmias or any arrhythmia: Cardiotrophin-1, Growth differentiation factor 15, Annexin A5, and serum microRNA-27b. If there is no evidence of the association between the biomarker and arrhythmias in hypertrophied ventricles, the biomarkers should be removed from this review or relocated to a section of potential biomarkers to investigate in the future.

Similar considerations apply to the soluble suppression of tumourigenicity-2, which is associated with arrhythmogenic right ventricular cardiomyopathy.

Midregional pro-atrial natriuretic peptide and apelin are related to atrial fibrillation, but no relation to ventricular arrhythmias has been found yet.

Recent references regarding annexin 5 should be included.

Minor comments

Line 70, Trypanosoma cruzi should be in italics

Line 71, remove “the” before Chagas

Line 189, add B-type natriuretic peptide before the acronym NT-proBNP.

Line 205, add high-sensitivity cardiac troponin I and high-sensitivity cardiac troponin T before the acronyms Hs-cTnI and hs-cTnT, respectively.

Line 224, C-reactive protein is first mentioned in the text.

Line 224, Left ventricular mass index is first mentioned in the text.

Line 314, add Matrix metalloproteinases and tissue inhibitor of matrix metalloproteinases before the abbreviations MMPs and TIMP, respectively.

Line 321/322, the correct spelling is tachyarrhythmia.

Reviewer 2 Report

The manuscript is scholarly and has subject matters pertinent to IJMS audience and clinicians.

Author Response

REVIEWER 2

The manuscript is scholarly and has subject matters pertinent to IJMS audience and clinicians.

Thank you for this evaluation.

Reviewer 3 Report

Dear authors,

The conception of your paper is nice and interesting. However, the manuscript should be extensively revised and rewritten in order to improve English language. Please, pay more attention to the style. For example, very often you abbreviate an item that you cite for the first time, I was lost in many points as I was unable to understand these abbreviations.

I think that an extensive revision is warranted before to consider this manuscript worthy for publication.

Please find some suggestions, below:

Line 12 : LVH concern only the left ventricle and not "an organ";

Lines 16-18: redundant, obvious, please delete or rephrase;

Line 19: rephrase, please do not repeat twice the same word;

Line 30: please define hyperplasia as well, it could be useful to the reader to know both definitions;

Lines 31-34: It would be better to speak about aetiology (i.e. secondary LVH....) rather than associated conditions. For example is myocardial infarction a cause or a consequence of LVH? As you state in this sentence it appears as a cause, but it is known that it could be a consequence of LVH as well;

Lines 40-41: Sorry you persist in make confusion if you list causes and consequences in the same sentence. Or is just a language issue? It is not clear to me;

Line 46: the graph is not clear at all. How "biomarkers" is a main group and also a subgroup of this main group??;

Line 83: LVH;

Line 85: Please the abbreviation should put in brackets the first time you mention LV mass, than you use only abbreviation;

Lines 103-105: a verb is lacking in the sentence;

Line 106: the are ????;

Line 170: what = amplitude?;

Line 175: If you speak about ECG predictors of SCD you should cite also other ECG or Holter features (e.g. R on T phenomenon etc...) and patterns of arrhythmia organization and complexity;

Line 189; please cite the physiopathological stimulus to BNP incretion;

Lines 214, 229, 236, 261…: do not use abbreviation the first time you mention an item;

Lines 240-241: it does not make sense;

Conclusion is poor, not based on clinical or scientific evidences.

Round 2

Reviewer 1 Report

The authors made major changes to the original version. The improvement is evident, and I recommend accepting the review in the present form.

Reviewer 3 Report

Thank you for changes, they correspond to my suggestions and justified when in disagreement.